# Specialised Surgical Instruments for Endoscopic and Endoscope-Assisted Neurosurgery: A Systematic Review of Safety, Efficacy and Usability

**DOI:** 10.3390/cancers14122931

**Published:** 2022-06-14

**Authors:** Holly Aylmore, Emmanouil Dimitrakakis, Joshua Carmichael, Danyal Z. Khan, Danail Stoyanov, Neil L. Dorward, Hani J. Marcus

**Affiliations:** 1UCL Queen Square Institute of Neurology, University College London (UCL), London WC1N 3BG, UK; d.khan@ucl.ac.uk (D.Z.K.); h.marcus@ucl.ac.uk (H.J.M.); 2Wellcome/EPSRC Centre for Surgical and Interventional Sciences (WEISS), University College London (UCL), London W1W 7TY, UK; e.dimitrakakis@ucl.ac.uk (E.D.); danail.stoyanov@ucl.ac.uk (D.S.); 3Department of Renal Medicine, Centre for Genetics and Genomics, University College London (UCL), London NW3 2PF, UK; j.carmichael.16@ucl.ac.uk; 4National Hospital for Neurology and Neurosurgery, London WC1N 3BG, UK; neil.dorward@nhs.net

**Keywords:** endoscopy, cancer, keyhole surgery, surgical instruments

## Abstract

**Simple Summary:**

The greatest technical barrier in endoscopic and endoscope-assisted neurosurgery is the instruments employed for these approaches. This systematic review aimed to identify specialised instruments for this type of surgery and evaluate their safety, efficacy and usability. We identified 50 instruments over 60 studies that were broadly safe and effective and generally considered to be ergonomic, though the learning curve was often noted as a disadvantage. Only eight studies compared the new instrument to standard instruments and comparisons were generally favourable to the new instrument. The development of novel and specialised instruments for endoscopic and endoscope-assisted neurosurgery is an area of interest for the field, but these instruments do not meet the need for improved articulation and future development should be based on established guidelines for neurosurgical innovation.

**Abstract:**

While there have been great strides in endoscopic and endoscope-assisted neurosurgical approaches, particularly in the treatment of deep-sited brain and skull base tumours, the greatest technical barrier to their adoption has been the availability of suitable surgical instruments. This systematic review seeks to identify specialised instruments for these approaches and evaluate their safety, efficacy and usability. Conducted in accordance with the PRISMA guidelines, Medline, Embase, CENTRAL, SCOPUS and Web of Science were searched. Original research studies that reported the use of specialised mechanical instruments that manipulate tissue in human patients, cadavers or surgical models were included. The results identified 50 specialised instruments over 62 studies. Objective measures of safety were reported in 32 out of 62 studies, and 20 reported objective measures of efficacy. Instruments were broadly safe and effective with one instrument malfunction noted. Measures of usability were reported in 15 studies, with seven reporting on ergonomics and eight on the instruments learning curve. Instruments with reports on usability were generally considered to be ergonomic, though learning curve was often considered a disadvantage. Comparisons to standard instruments were made in eight studies and were generally favourable. While there are many specialised instruments for endoscopic and endoscope-assisted neurosurgery available, the evidence for their safety, efficacy and usability is limited with non-standardised reporting and few comparative studies to standard instruments. Future innovation should be tailored to unmet clinical needs, and evaluation guided by structured development processes.

## 1. Introduction

In Lewis Carroll’s 1871 novel Through The Looking Glass, and What Alice Found There, Alice falls into a new world that shifts her perception of what she thought she knew and was possible [1]. Over the past 100 years neurosurgeons have been through their own looking glass: the endoscope. In the last 30 years, many new endoscopic and endoscope-assisted neurosurgical approaches and procedures have been developed, facilitated by a rise in complimentary innovation in endoscopic instruments and surgical techniques [2].

The development of endoscopic and endoscope-assisted approaches has played a key role in treating and improving the outcomes of brain tumours. The endoscopic endonasal transsphenoidal approach, for instance, may decrease the incidence of surgical complications when compared with traditional microsurgical cases in treating pituitary adenomas, likely a direct result of the improved visualisation of anatomy this approach provides [3]. Utilisation of endoscopic approaches in treating patients with sinonasal and ventral skull base cancers has also been found to significantly improve patient quality of life scores within the first postoperative year [4].

However, despite the great strides made in endoscopy, it is the use of surgical instruments in endoscopic approaches that has remained the greatest technical barrier to their adoption [5]. Presently, many of the instruments used in endoscopic and endoscope-assisted neurosurgical approaches have been adopted from the armamentarium of neighbouring specialties such as rhinology and urology. These include instruments that grasp, such as forceps, and instruments that cut and divide, such as scissors, knives and punches [6]. Although most such instruments are straight, some angled instruments are also in popular use, particularly in combination with angled endoscopes. Alongside these traditional instruments special classes have been developed, including microdebriders for precise tissue removal, ultrasonic devices for removal of firm tumours that are not easily suctioned and mono- and bipolar forceps for coagulation.

Historically, the introduction of new instruments such as these has been unstructured and variable. Though assessment frameworks for innovation in neurosurgery, including the development of novel neurosurgical instruments, have been developed [7,8], an instrument may be used for the first time in the form of a dedicated research study but, more often, may be published as a non-comparative trial without special institutional board review. Although many such instruments are safe and effective, the dangers of this process are obvious and have been frequently reported in the literature [9]. The usability of a new instrument, including its ergonomics and learning curve, is also an important factor when considering its adoption within the surgical community.

The aims of this review were therefore first, to identify specialised instruments that have been developed for endoscopic and endoscope-assisted neurosurgery; second, to assess the evidence for the safety and efficacy of these instruments; and third, to assess their usability.

## 2. Materials and Methods

This review was conducted and the results are presented in accordance with the Preferred Reporting Items for Systematic Reviews and Meta-Analyses (PRISMA) statement [10]. The review protocol was registered on the PROSPERO database (CRD42020209146).

### 2.1. Literature Search

A search of the Cochrane Controlled Register of Trials (CENTRAL), MEDLINE (via Ovid), Embase (via Ovid), SCOPUS and Web of Science databases from their inception was performed on 15 June 2020. No filters or limitations were applied to the search. Keywords, Medical Subject Heading (MeSH) terms and their synonyms for “surgical instrument”, “endoscopic surgery” and “neurosurgery” were combined. The full search strategies for each database are available in Appendix B. Following screening and selection, an additional manual search of the reference lists of included studies was conducted.

### 2.2. Eligibility Criteria

The following ‘PICO’ framework was employed to assess the eligibility of articles: Population: human patients, surgical models, cadavers; Intervention: specialised instruments for endoscopic or endoscope- assisted neurosurgery; Comparison: standard instruments for neurosurgery; Outcome: safety measures (blood loss, intraoperative complications, postoperative complications, mortality rate), efficacy measures (EOR, operative time), assessments of ergonomics, assessments of learning curve. Eligible articles were included in the review if they met the following criteria: the specialised instrument is mechanical with the aim of manipulating tissue; the instrument was applied to endoscopic or endoscope-assisted neurosurgery; and the study was an original research paper, including preclinical and clinical studies.

Eligible articles were excluded if they met any of the following criteria: the study evaluated a device or system that was not a mechanical instrument with the aim of manipulating tissue; the instrument was not applied to endoscopic or endoscope-assisted neurosurgery; or the study was not an original research paper.

### 2.3. Study Screening and Selection

Results of the final searches of all databases were imported to EndNote X9.3.2 (Philadelphia, PA, USA). After removal of duplicates, an initial screening of the title and abstract of each article using the eligibility criteria was performed. The full text of articles that remained included following this screen were then read to ensure all eligibility, inclusion and exclusion criteria were met before extracting data. All stages of screening were carried out independently by two reviewers (H.A. and J.C.). Any discrepancies between the two screenings were discussed and resolved with a third reviewer (H.J.M.).

### 2.4. Data Extraction and Analysis

The following data were extracted from included articles: study design; study population and sample size; pathology; procedure; instrument name; company that produced the instrument and their location; type of surgery the instrument was for; instrument features; instrument function; reports of instrument malfunction; patient outcomes measures that provided evidence for safety; general comments on safety; surgical outcomes that provided evidence for efficacy; general comments on efficacy; any comparison to standard neurosurgical instruments and which instruments; any ergonomic assessment or comments on instrument ergonomics; any learning curve assessment or comment on instrument learning curve; and limitations of the instrument. Summary statistics with accompanying narrative synthesis were generated for the identified available instruments by safety, efficacy and usability metrics used. Comparative statistical analyses were not performed due to the heterogeneity of the underlying metrics.

## 3. Results

### 3.1. Search Results

Following the search strategy detailed in Section 2 and Appendix B, a total of 5757 articles were identified (Figure 1). After removal of duplicates 4143 titles and abstracts were screened. Following the full-text screen, 62 articles were identified as fully meeting the eligibility and inclusion criteria and were included in the final analysis. The reference lists of these 62 articles were then screened for other potentially relevant articles that were not identified through the database search. This screening yielded zero results.

Most papers, 78% (48/62 studies), were case series. An additional 11% (7/62) were reports of a novel instruments design, 5% (3/62) were controlled trials, 3% (2/62) were single case reports and 3% (2/62) were cadaver or model studies. Most studies were conducted by teams in the USA, 31% (19/62), though all continents were represented in the included studies. There is a steep rise in included papers published after 2010 (Figure 2).

Novel instruments in included studies were applied to a variety of specialities (Figure 3). The most common application of a novel instrument was for the treatment of pituitary tumours using the endoscopic endonasal transsphenoidal approach (eight instruments), followed by endoscopic third ventriculostomy for hydrocephalus (seven instruments).

### 3.2. Available Instruments

A total of 50 specialised instruments for endoscopic or endoscope-assisted neurosurgery were identified (Table 1). Most available specialised instruments were for general use in all endoscopic and endoscope-assisted neurosurgical approaches, but served a single function. The most common function of these instruments was resection (28 instruments), followed by dissection (5), coagulation (5), electrocautery (4) and retraction (4) (Figure 4).

Seven available instruments, the XS Micro Instruments, Modified Suction Tip, Marburg Electrosurgical Probe, Harmonic Scalpel, Bipolar Microforceps, Bipolar Microscissors and the Handpiece, Keyhold, and Needle-Type Probes and Probe Sheaths for use with the Ultrasonic Surgical Unit were reported for general use with multiple functions.

An additional nine instruments were reported for specific approaches or procedures and capable of performing specialised single-functions: two instruments were designed for endoscopic third ventriculostomy to dilate the floor of the third ventricle; one instrument for craniosynostosis surgery; one as suction for intraventricular haematoma procedures; one to retrieve resected sections of colloid cysts; one for retraction during laminectomy; one for retraction during the endoscopic endonasal transsphenoidal approach; one for severing the sympathetic nerve of thoracic vertebrae; and one as a cannula for resection instruments.

### 3.3. Evidence for Safety

Reporting of evidence for the safety of available instruments is varied with non-standardised outcomes, with 11 studies not reporting on safety at all (Appendix A).

An additional 19 out of 62 studies only provide a general comment with no objective measures such as “according to our experience in more than 100 applications this bipolar instrument is safe” [16] or report comments on objective measurements such as a decrease in operative blood loss but do not report data [59].

The highest rate of intraoperative complications is 23%, reported for the BoneScalpel and Piezoelectric System in 13 patients with craniosynostosis undergoing endoscope-assisted surgery [18]. In this study the intraoperative complication rate is combined for the use of both instruments so it is not possible to accurately assess the complication rate of each individually. The intraoperative complication rate for the NeuroBalloon [43] also cannot be calculated as the authors report “more than 1000 cases” with no definitive number given.

The highest post-operative complication rate is 50% reported for the Micro ENP Ultrasonic Handpiece in two cases of endoscopic third ventriculostomy for intraventricular tumours [36]. However, it is unclear if this complication, ventriculitis, is directly related to the instrument or the operation given that ventriculitis is a known complication [73]. The only other instruments to report a postoperative complication rate over 10% were the 6.3 mm Percutaneous Endoscopic Instrument [13] utilised to treat migrated disk herniation in 22 adult patients, the 980 nm Diode Laser [14] utilised in nine adult patients with hydrocephalus, the OmniGuide CO2 laser [55] utilised in a case series of 16 adult patients with pituitary lesions that underwent surgery using the endoscopic endonasal transsphenoidal approach and the SONOCA Ultrasonic Aspirator [63] utilised in nine adult patients with supratentorial intraventricular tumours that underwent endoscopic resection.

### 3.4. Evidence for Efficacy

Reporting of the efficacy of available instruments is also varied and non-standardised (Appendix A). There is no reporting on efficacy in nine out of 62 studies and 33 only report a general comment such as “proven to be a valid and effective instrument” [41] with no outcome measures.

Mean operative time was reported in 10 studies, ranging from less than 10 min for the Artemis Neuro Evacuation Device for the treatment of a large suprasellar tumour with cavernous sinus invasion using the endoscopic endonasal approach [15] to 114 min for the Bipolar Microscissors, the mean operative time over 100 cases of intracranial astrocytoma treated with neuroendoscopic surgeries [17]. EOR, either gross, subtotal or partial was also reported in nine studies with GTR ranging from 33.33% for the SONOCA Ultrasonic Aspirator using endoscopic approaches for intraventricular tumours [62] to 100% for both the Self-Retaining Retractor used for the endoscopic endonasal transsphenoidal approach in pituitary adenomas [60] and the Sonopet Ultrasonic Bone Aspirator when operating on skull base tumours using the endoscopic endonasal approach [67].

### 3.5. Evidence for Usability

#### 3.5.1. Ergonomic Assessment

Overall, only 7 out of 62 studies report outcomes or comments relating to the ergonomics of the instrument (Appendix A). Comment on instrument ergonomics featured in six studies of six instruments: the Suction Device made of Shape Memory Alloy connected to the ATOM5 Record 55 DDS [68], the Malleable Endoscope Suction Instrument [34], the Monoshaft Bipolar Cautery [42], the Bipolar Microscissors [17], the Sonopet Ultrasonic Bone Aspirator [66] and the Handpiece, Keyhold and Needle-Type Probes and Probe Sheaths for use with the Ultrasonic Surgical Unit [31]. One study reported administering a seven-question survey in which surgeons rated the ergonomics of the final instrument design as “very good” [34].

#### 3.5.2. Learning Curve Assessment

Comments on the learning curve of the available instruments were reported in eight out of 62 studies (Appendix A). The learning curve is reported to be improved or similar to standard instruments for the 2.0 μm Diode Pumped Solid State (DPSS) Laser, “steep and quick” [12], the Bipolar Microscissors, “no special training is needed” [17], NeuroBalloon, “use of the NeuroBalloon catheter in our experience may shorten this learning process” [43], the Sonopet Ultrasonic Bone Aspirator, “there did not appear to be a difference in learning curve” [64] and the ZESSYS, “could reduce the technique difficulties” [72].

A study of the NICO Myriad reports the learning curve as a disadvantage [46], however they note this is the case for all devices. Though the learning curve is commented on for the SONOCA Ultrasonic Bone Aspirator, the comment pertains to the difficulty of the surgical technique rather than the use of the instrument [63].

### 3.6. Comparison to Standard Instruments

Comparison to standard neurosurgical instruments was part of the study design of eight out of 62 studies for five instruments (Appendix A). Kawamata, Amano, and Hori [24] comment that the Flexible Forceps used during endoscopic endonasal approaches for pituitary tumours were “able to access sites where regular dedicated instruments... could not easily reach” facilitating improved manipulation of tissue in comparison to standard excision instruments, while Kawamata and colleagues [30] report that the Harmonic Scalpel for dissection of intracranial tumours demonstrates a “much lower temperature and much smaller area of elevated temperature than monopolar or bipolar diathermy” devices. Additionally, Nakamura et al. [44] comment that the “clinical and radiological results in the NAC [New Angled Chisel] group were better” in patients undergoing the procedure with the novel instrument in comparison to patients who underwent the same procedure without the NAC. These three studies do not report any objective outcomes.

The Bipolar Microscissors used to treat intracranial astrocytomas show reduced mean blood loss and operative time in comparison to standard instruments, though whether or not the difference is significant is not reported (361 mL vs. 278 mL and 145 min vs. 114 min) [17]. Mean operative time was improved when comparing a standard cylindrical tubular retractor to the Novel Rectangular Tubular Retractor [54], standard instruments to the Piezoelectric System [56], and TESSYS instrument to the novel ZESSYS [72]. The Modified Neuroendoscope Technology (MNT): a transparent sheath and haematoma smashing aspirator utilised for endoscopic evacuation of cerebral haemorrhages reduced mean ICU monitoring time (5 days vs. 16.2 days, *p* = 0.000), improved 6-month mortality rate (6.7% vs. 22.5%, *p* = 0.036) and improved 6-month Glasgow Outcome Scale result (3.7 vs. 2.9, *p* = 0.021) in comparison to procedures using an existing external drainage and monitoring system [38]. Mean operative time increased (86.6 min vs. 39.7 min, *p* = 0.000), though given the improved safety markers this does not appear to be a significant limitation. Similarly, the Sonopet Ultrasonic Bone Aspirator improved mean blood loss (16.55 mL vs. 22.58 mL, *p* = 0.000) and mean operative time (31.92 min vs. 41.33 min, *p* = 0.000) in comparison to standard Jansen–Middleton kerrison, thru-cut and backbiter instruments for endoscopically resected supratentorial intraventricular tumours [64].

## 4. Discussion

### 4.1. Principal Findings

This systematic review identified 50 specialised instruments for endoscopic and endoscope-assisted neurosurgery across 62 studies. Notably, there was a steep rise in included papers published in the last decade.

The novelty of development in almost all instruments was in what they do, such as a new way to resect a skull base tumour, rather than in how they do it, i.e., the articulation of the instrument. The evidence for the safety and efficacy of these instruments was limited with non-standardised reporting across studies. Very few studies reported on usability assessments and in those that did all but one reported only subjective comments with no objective measures.

The development of specialised instruments for endoscopic and endoscope-assisted neurosurgery is a growing area fuelled by technological innovation. However, the innovation demonstrated in the identified studies within this review does not solve the established issues with standard instruments: namely, issues of triangulation [74] and articulation that would enhance flexibility and mobility of the instrument [75].

### 4.2. Safety

It is essential that the field finds a way to evaluate safety objectively in order to demonstrate that novel instruments have merit in their use and that they are an improvement on currently available instruments. This comparison is the “gold standard” in pharmacology and with adoption of the IDEAL-D and IDEAL frameworks [7,8] could become the gold standard for the development of surgical instruments in endoscopic neurosurgery. While this lack of comparison exists it is difficult to establish if the outcome measures reported for safety are a result of the instrument causing improvement or deterioration, or if this is a result of the pathology or surgical approach.

### 4.3. Comparison with Other Studies

Our review found that no studies used either an already established structured framework for any assessment of a novel instrument or devised a new one. While the development of such frameworks is a recent advance in improving the design and evaluation process for surgical instruments, over half of the studies identified were published after the design criteria for instruments for endoscopic neurosurgery had been reported [47,75] and just over a third of studies after publication of the IDEAL framework, developed specifically for neurosurgical instruments, in 2013 [7].

Our findings are in keeping with the wider literature on neurosurgical innovation. A recent systematic review of the use of the IDEAL framework in innovation for the endoscopic endonasal approach for skull base meningiomas identified 26 studies, none of which could be classified on the IDEAL framework, with some first-in-man studies being reported before pre-clinical studies of the same instrument [76]. However, there is some data to suggest that the IDEAL and similar frameworks are being adopted, albeit slowly, within the neurosurgical community. In a recent bibliometric analysis Ota et al. found 51 studies that cited IDEAL with the overwhelming majority being positive [77].

### 4.4. Strengths and Limitations

The most significant limitation of this review is that the nature of the literature required the pooling of results of hugely disparate outcomes. This invites bias in the interpretation and conclusions drawn, particularly as it was not possible to conduct comparative statistical tests of quantitative measures such as mean blood loss or operative duration due to the small sample size of studies that reported the same objective measures. However, this review was systematic and exhaustive. The search of the literature was broad, with no limiters placed on the databases, in order to ensure all the relevant literature was considered for inclusion based upon pre-established and clear criteria.

## 5. Conclusions

While there are many specialised instruments for endoscopic and endoscope-assisted neurosurgery available, the reporting of their safety and efficacy is limited. Specialised instruments are infrequently compared to standard instruments, meaning accurate conclusions to establish if they provide improved, cost-effective, surgical options for patients cannot be drawn. The usability of these instruments is also not regularly or objectively assessed, raising concerns about the adoption of new technologies and the effect of these measures on safety and efficacy.

This review shows that novel specialised instruments for endoscopic and endoscope-assisted neurosurgery are an area of interest for the field with many devices being developed in the past few years to overcome procedural issues with standard instruments that make them sub-optimal for these approaches. Despite the increase in new instruments, almost all of the identified devices are novel in what they do rather than in how they do it, mostly innovating the end effectors of the instrument but not the unmet need for improved articulation.

Available instruments define the applications and limitations of endoscopic and endoscope-assisted neurosurgery. Given that the identifed studies are inconsistent in their reporting of safety, efficacy and usability future research should seek to establish guidelines for surgical innovation, developing the IDEAL and IDEAL-D frameworks towards a more practical criteria that can be applied successfully in future studies of specialised instruments.

## Figures and Tables

**Figure 1 cancers-14-02931-f001:**
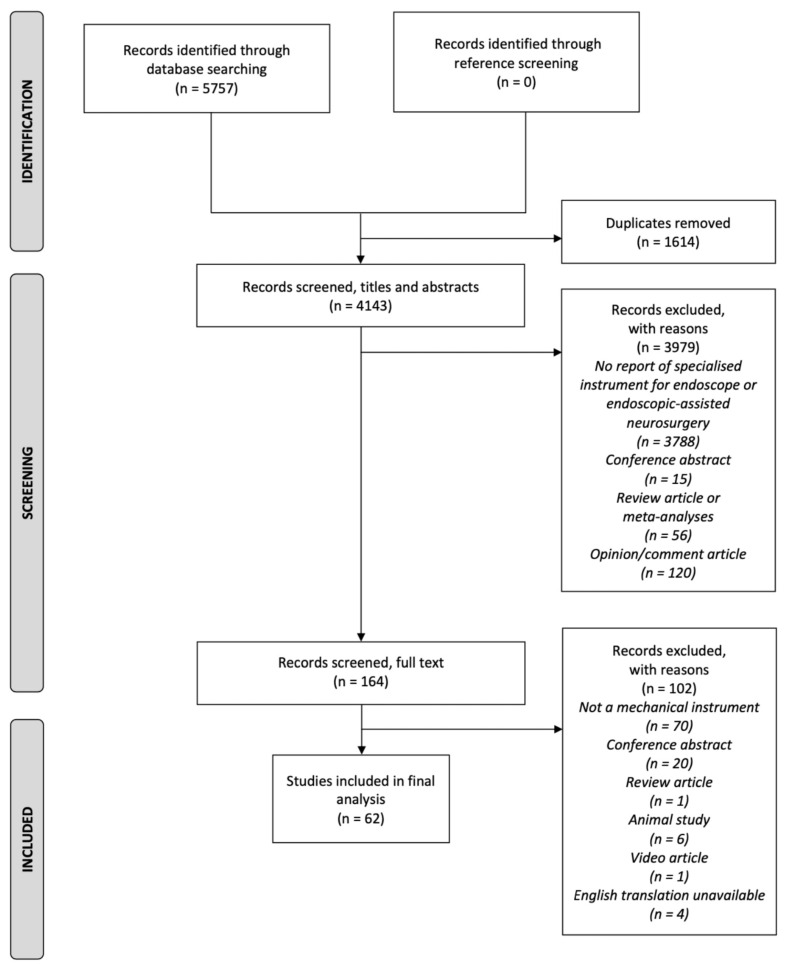
PRISMA flow diagram of the inclusion and exclusion of studies.

**Figure 2 cancers-14-02931-f002:**
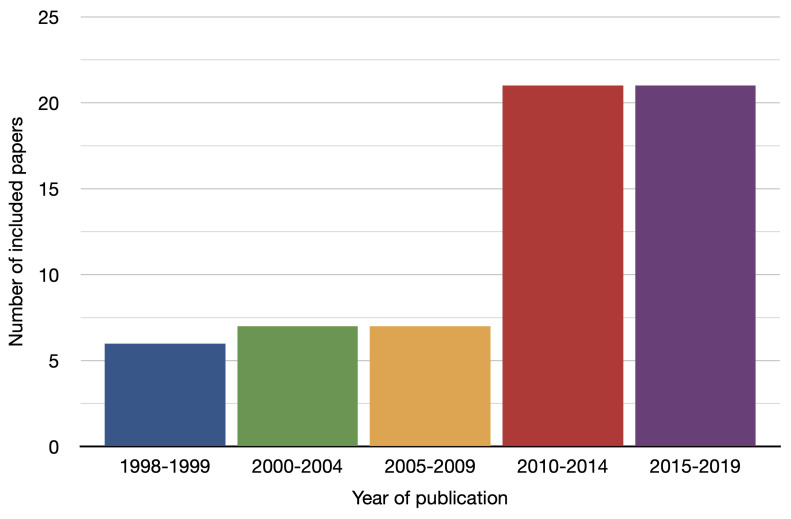
Year of publication of included studies.

**Figure 3 cancers-14-02931-f003:**
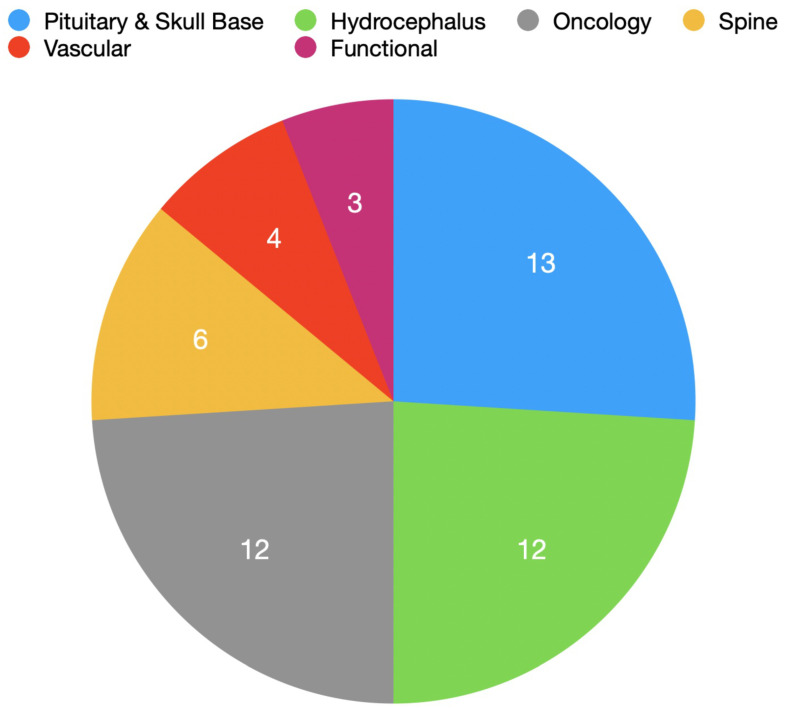
Number of novel instruments identified by neurosurgical speciality.

**Figure 4 cancers-14-02931-f004:**
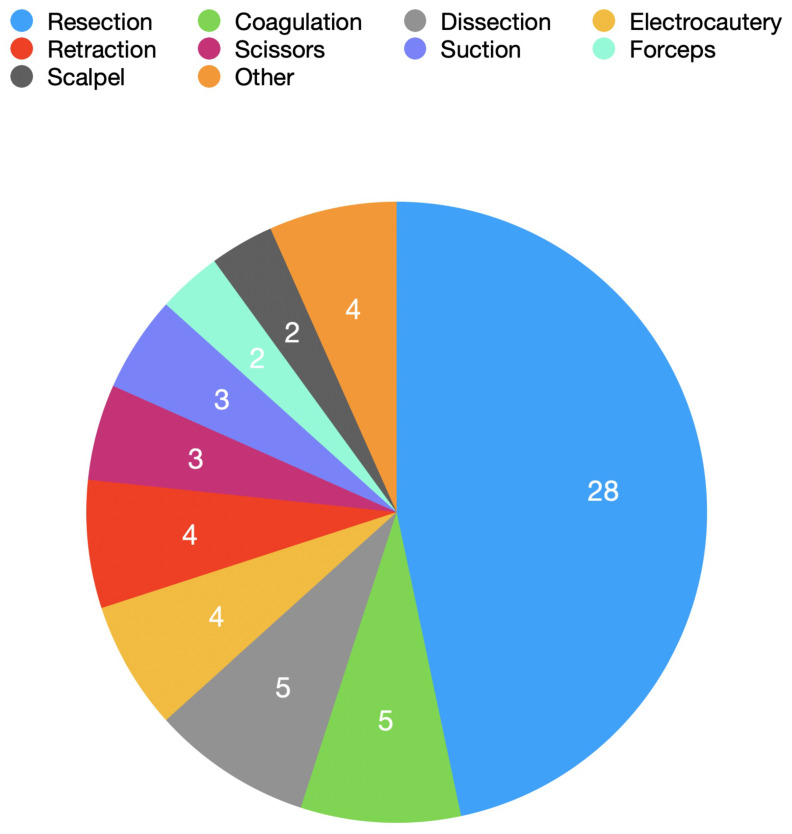
Number of novel instruments identified by function of instrument.

**Table 1 cancers-14-02931-t001:** Summary of available instruments for endoscopic or endoscope-assisted neurosurgery.

Name and Company	Type	Function	Features	Identified Studies (Author, Year)
2.0 μm Diode Pumped Solid State (DPSS) Laser, (RevoLix, LISA laser products, Katienburg, Germany)	General Purpose	Coagulation	Avoids tissue disruption due to rapid formation and collapse of steam bubbles, precise and atraumatic tissue opening of approximately 1 mm diameter can be achieved	Ludwig, 2007 [11]; Schuhmann, 2009 [12]
6.3 mm Percutaneous Endoscopic Instrument, (ASAP, Umkirch, Germany)	General purpose	Resection	One thin and slightly curved jaw with sawtooth surface and concave part that rotates in perpendicular plane to the mouth opening	Nakamura, 2019 [13]
980 nm Diode Laser, (MediLaser 980 nm, DMC Equipamentos LTDA)	General purpose	Dissection	Provides proper absorption in both water and haemoglobin, maximum continuous wave output power of 25 W, pulsed mode with frequency operation interval ranging from 0.16 to 1 kHz and pulse widths ranging from 0.5 m s to 6 s, single-pulse mode available	Reis, 2016 [14]
Artemis Neuro Evacuation Device, (Penumbra, Alameda, CA, USA)	General purpose	Resection	Long narrow “wand” shaft similar to 9-French suction, allows free movement of tip	Yu, 2009 [15]
Bipolar Microforceps, (Erbe GmbH, Tubigen, Germany)	General purpose	Bipolar electrocautery, Forceps	Outer diameter 1.5 mm, length 360 mm, branches open smoothly to width of 6 mm, can be used for grasping/spreading tissue and bipolar coagulation	Riegel, 2002 [16]
Bipolar Microscissors, (Authors own)	General purpose	Bipolar electrocautery, Cutting—Scissors	Pistol-shaped microscissors combined with a lead consisting of two poles of a bipolar coagulator, two 10 mm blades	Qiu, 2004 [17]
BoneScalpel, (Misonix, Farmingdale, NY, USA)	Crainosynostosis	Resection	Ability to section bone precisely while sparing soft tissue	Chaichana, 2013 [18]
Calvian Endo-Pen, (Sutter Medizintechnik, Freiburg, Germany)	General purpose	Coagulation	Long straight bipolar instrument with slightly angled fine and thin tips that can be closed by compression of hand piece, angled working tips designed to enhance visibility of tip and coagulation zone in situ	Gerlach, 2018 [19]
Chole-Dacey Transnasal Transsphenoidal Speculum, (Anspach, USA)	General purpose	Retraction	Possible to convert between endoscopic and microscopic approach during procedure, interchangeable stainless steel anodized aluminium blades, can be constructed of nonferromagnetic materials for using during intraoperative MRI	Chole, 2011 [20]
EASYTRAC, (Walnut Medical, AIIMS, Delhi, India)	General purpose	Retraction	V-shaped compressed construction, three sizes (paediatric/small/large adult), SS-titanium alloy, 0.5 mm thick, non-reflective black coating	Chandra, 2019 [21]
Endoscopic Curved Kerrison Rongeur, (Authors own)	General purpose	Resection	Modified curved Kerrison rongeur with channel to fit small malleable endoscope, small enough to be readily inserted and used within lateral recess	Frank, Martin, & Hsu, 2002 [22]
Endoscopic Stenosis Retractor, (Authors own)	Laminectomy	Retraction	When retracting the dura instrument permits simultaneous visualisation of anatomy of lateral recess and the activity of instruments used to decompress it	Frank & Hsu, 2002 [23]
Flexible Forceps, (Authors own)	General purpose	Resection	Forceps portion is 3 mm in diameter, total length 335 mm, working length 240 mm, flexible tip 10 mm long	Kawamata, 2008 [24]
Guillotine Knife, (Authors own)	General purpose	Cutting —Scalpel	Designed to be used with Lotta ventriculoscope, outer diameter 2/7 mm, working length 30 mm, consists of shaft, blade and handle, cutting mechanism operates on principle of a guillotine with sharp downward-moving blade that slides into groove within footplate and cuts tissue on edge of groove	El Damaty, 2014 [25]
Gyrus Diego Microdebrider, (Gyrus ACMI-ENT Division, Bartlett, TN, USA)	General pupose	Resection	Suction-based powered instrument with a blunt end that consists of a hollow outer shaft with an inner rotating motor that can be attached to different blades at opening of distal tip	Patel, 2014 [26]
Haemostatic Agent Delivery, (Authors own)	General purpose	Delivery of haemostatic agent	2 tubes, internal tube connected to suction and external tube, internal tube introduced to proximal end of external tube, any conventional haemostatic agent is inserted or injected into distal end of external tube	Waran, 2011 [27]
Handpiece for SONOCA Ultrasonic Aspirator, (Sonoca, Söring, GmbH, Quickborn, Germany)	General purpose	Resection	Frequency range 20–80 kHz, vacuum suction 0–0.9 bar, allows precise and effective aspiration of tissue	Oertel, 2008 [28]
Handpiece, Keyhold, Needle-Type Probes, and Probe Sheaths for use with the Ultrasonic Surgical Unit, (Olympus Optical Co., Tokyo, Japan)	General purpose	Resection, Suction	Weight is 90 g, keyhole-type probes have 93 and 112 mm lengths with 2.2 mm tip diameter and 9.5 and 11.2 mm sheath diameters at most proximal side, needle-type probes have 89 and 171 mm lengths with 1.9 mm tip diameter and 3.5 and 3.3 mm sheath diameters at proximal side, all compatible with magnetic resonance imaging	Sawamura, 1999 [29]
Harmonic Scalpel, (Ethicon Endo-Surgery Inc., Cincinnati, OH, USA)	General purpose	Coagulation, Cutting—Scalpel	Vibration frequency 55,500 Hz, composed of generator, handpiece, and blade, cutting speed and coagulation can be adjusted	Kawamata, 2001 [30]
HelixFlex, (Authors own)	General purpose	Resection	Steerable tip that measures 5.8 mm in diameter and 60mm in length, each layer of cables has six stainless steel cables resulting in 18 cables that are fixed at free end of top, 5 guiding plates kept in place by springs and containing guiding holes for cables are placed along length of tip, centre contains flexible and axially incompressible tube	Gerboni, 2015 [31]
Helix Hydro-Jet, (Erbe GmbH, Tubigen, Germany)	General purpose	Dissection	Narrow nozzle that is 100 or 120 µm in diameter, through this instrument sterile 0.9% isotonic saline is emitted at pressures ranging from 1 to 150 bars	Oertel, 2006 [32]
Lotan’s Hook, (Authors own, manufactured by Contact Medical Ltd., Ramat Hasharon 47279, Tel Aviv, Israel)	Severance of the sympathetic nerve	Cutting	Metal hook, 43 mm long, shaft 5 mm thick, distal 3 cm of the device that contains the hook is 1.5 mm thick, hook placed at 120 degree angle to longitudinal shaft of device	Lotan, 2001 [33]
Malleable Endoscope Suction Instrument, (Authors own)	General purpose	Suction	Malleable 1.2 mm diameter channel soldered to side of 20 cm malleable 9-French suction	Frank & Ragel, 1998 [34]
Marburg Electrosurgical Probe, Bipolar, Flexible, (Authors own)	General purpose	Cutting, Coagulation	Single-use bipolar flexible needle in working channel of endoscope preventing mechanical resistance	Hellwig, 1999 [35]
Micro ENP Ultrasonic Handpiece, (Söring GmbH, Quickborn, Germany)	General purpose	Resection	Ultrasonic bone aspirator	Selvanathan, 2013 [36]
Modified Flexible Grasping Forceps, (Hopital Henri Mondor, Creteil, France, with assistance of Karl Storz Endoscope GmbH, Tuttlingen, Germany)	ETV	Dilation of the floor of third ventricle	Tip thin enough to allow perforation of floor of ventricle but blunt shape cannot damage structures as needle could, smooth inner surface of jaws, outer surface has indentations to catch edges of ventriculostomy preventing them from slipping away	Decq, 2000 [37]
Modified Neuroendoscope Technology (MNT): a transparent sheath and haematoma smashing aspirator, (Authors own)	Intraventricular haematoma	Suction	Transparent sheath with 7 mm outer diameter, haematoma smashing aspirator contains 3 mm spiral suction device	Du, 2018 [38]
Modified Nippon Medical School Type, (Fujita Ika, Tokyo, Japan)	General purpose	Suturing	Single-shaft 170 mm bayonet needle holder, tip bent upwards at 45 degree angle with nicks in both sides of lateral walls	Jimbo, 2013 [39]
Modified Suction Tip, (Authors own)	General purpose	Dissection, Resection	Redesigned tip making it more bulbous with longitudinal slices and blunt margins	Faraj, 2017 [40]
Monopolar Suction Cautery, (Authors own)	General purpose	Monopolar electrocautery	Surgical endoscope composes main body of suction cautery, intranasal portion of tube covered with rubber catheter, aspiration system connected with tube	Pagella, 2016 [41]
Monoshaft Bipolar Cautery, (Authors own)	General purpose	Bipolar electrocautery	Diameter 3 mm, can be manipulated inside a narrow endoscopic corridor, bipolar coagulation at 2 or 10 watts	Nagasaka, 2011 [42]
NeuroBalloon, (Integra LifeSciences Corp., Princeton, NJ, USA)	ETV	Dilation of the floor of third ventricle	4-F catheter with a double-barrel violin-shaped balloon at the distal end	Guzman, 2013 [43]
New Angled Chisel, (Authors own)	General purpose	Resection	Has 4 or 5 mm wide blade that is angled at 20 degrees, hammered portion located on the end of protruded branch that extends from bottom of grip on shaft, when force is applied on hammered portion it travels in direction of angled blade	Nakamura, 2017 [44]
NICO Myriad, (NICO Corporation, IN, USA)	General purpose	Resection	High speed oscillating sharp inner cannula contained in stationary outer cannula with direct side window at end of outer cannula that allows surgeon to push normal tissue away from cutting aperture	Albright, 2012 [45]; Dlouhy, 2011 [46]; Garcia-Navarro, 2011 [47]; Goodwin, 2015 [48]; McLaughlin, 2012 [49]; Mohanty, 2013 [50]
Nitinoil Stone Retrieval Basket, (Boston Scientific, Marlborough, MA, USA)	Resection of colloid cyst	Retrieval of resected intraventricular tumour	Basket shaped with 3-Fr Zero Tip, opens to outer diameter of about 16 mm	Schirmer, 2011 [51]
Novel Burr Hole Dilator, (S&B Corporation, Chiba, Japan)	General purpose	Expansion/extension of burr holes	27 mm long, sharp 10 mm blades on side, arc form on bottom to prevent dural tear, cordless handle	Kuge, 2019 [52]
Novel Dilator for the Pipeline Minimally Invasive Retractor System, (DePuy Spine, Raynham, MA, USA)	General purpose	Dilation	Holes in dilating probe allow egress of fluid and markings enable surgeon to determine depth of placement, tip is smaller to decrease trauma to tissue	Dorman, 2008 [53]
Novel Rectangular Tubular Retractor, (Authors own)	General purpose	Retraction	An almost rectangular tube, cranial and caudal sides have curved surfaces to maintain length of retractor to 16 mm, upper part has cylindrical retractor, dilators are plates inserted into one side of spinous process	Nakamura, 2017 [54]
OmniGuide CO2 Laser, (OmniGuide, Cambridge, MA, USA)	General purpose	Dissection	Continuous-wave laser energy allows accurate cutting, ablation, and microcoagulation by using focused beams without excessive need to manipulate tissue, flexible-fibre allows access to narrow corridors	Jayarao, 2019 [55]
Pizeoelectric System, (Synthes, Inc., West Chester, PA, USA)	Crainosynostosis	Resection	Functional frequencies of 25 to 39 kHz, tips are available as blades and diamond bits including an angled cutting tip of 45 degrees, ability to section bone precisely while sparing soft tissue	Chaichana, 2013 [18]; Gellner, 2017 [56]; Mancini, 2012 [57]
Pulse Laser-Induced Liquid Jet, (Sparkling Photon, Inc., Tokyo, Japan)	General purpose	Dissection	Bayonet-shaped catheter incorporating a jet generator made of a stainless steel tube and optical quartz fibre leading to 19 G stainless tube with metal nozzle	Nakagawa, 2015 [58]; Ogawa, 2011 [59]
Self-Retaining Retractor, (Authors own, made from strip of polypropylene (Essiz Orthodonic plate, Raintree Essix Inc., Sarasota, FL, USA))	Elevation of redundant diaphragma	Retraction	Transparent flexible material self-retaining retractor tailored to adequate width of sellar opening	Kutlay, 2013 [60]
Series of Tipped Instruments: ring curettes, dissectors, hooks, pimer, (Croma Gio. Batta, Padova, Italy)	General purpose	Resection	Secure grip, barycenter of the instrument is the surgeon’s hands, elimination of bayonet-like shape with handle bent in horizontal plane to avoid interference with hands and allow distal thin part to be utilised	Cappabianca, 1999 [61]
SONOCA Ultrasonic Aspirator, (Sonoca, Söring, GmbH, Quickborn, Germany)	General purpose	Resection	Frequency range 20–80 kHz, vacuum suction 0–0.9 bar	Cinalli, 2017 [62]; Ibanez-Botella, 2019 [63]
Sonopet Ultrasonic Bone Aspirator, (Stryker, Kalamazoo, MI, USA)	General purpose	Resection	Ultrasonic oscilation to emulsify bone limiting damage to surrounding tissue, simultaneous irrigation and aspiration functions, different disposable tips	Baddour, 2013 [64]; Kim, 2006 [65]; Ledderose, 2019 [66]; Rastelli, 2014 [67]
Suction Device made of Shape Memory Alloy connected to ATOM5 Record 55 DDS, (Authors own)	General purpose	Suction	Cannula manufactured with a shape-memory alloy, can be adapted to a patients’ anatomy simply by bending it by hand during surgery, length of tube 150 mm	Grunert, 2018 [68]
Trapezoidal Specula, (Mizuho-America Inc. Beverly, MA, USA)	General purpose	Retraction	Working length of 60 mm, proximal 20 mm oval-shaped segment to conform to nostril shape, middle 20 mm segment has ventrically oriented blades, distal 20 mm segment transitions to trapezoidal orientation with distal blades angled 15 degrees outward and upward or downward depending on speculum used	Fatemi, 2008 [69]
Ultrasonic Aspirator Tube, (In cooperation with Olympus Optical Company, Tokyo, Japan)	General purpose	Resection	Outer diameter 1.8 mm and power of 15–30 watts, metal tip of aspirator made to vibrate at frequency of 24 kHz, can fragment and aspirate tissue simultaneously, maximal vacuum pressure around 600 mm Hg, on/off controls of irrigation and vibration via foot switch	Oka, 1999 [70]
XS Micro Instruments, (Aesculap BBraun, Tuttlingen, Germany)	General purpose	Forceps, Cutting—Scissors	Coaxial shaft, length varies between 6–10 mm, 2 and 3 mm diameter shafts	Cristante, 1999 [71]
ZESSYS, (Authors own)	Targeted foraminoplasty	Cannula for resection instruments	Dual-cannula adjustment with thinner cannula containing guide rod/K-wire, and larger cannula for bony abrasion by trephine/bone drill	Ao, 2018 [72]

## Data Availability

Not applicable.

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
