# Peer review of "Specialised Surgical Instruments for Endoscopic and Endoscope-Assisted Neurosurgery: A Systematic Review of Safety, Efficacy and Usability"

_cancers, 2022, doi:10.3390/cancers14122931_

Round 1
Reviewer 1 Report
I have no significant comments for this review paper. The aims of the review are clearly stated and are met as detailed in the body of the paper and conclusions. The methodology chosen is well structured and logical. References are chosen based on the method stated, providing an exhaustive review of the safety, usability and application of endoscopic instruments for neurosurgery. Happy for the review to be published as is.
Author Response
Thank you for taking the time to review our submission and for your comments.
Reviewer comment: English language and style are fine/minor spell check required.
Revisions made: A thorough proof-read of language has been performed and small changes made throughout.
If have you have any other comments following this revision, we would be happy to address them.
Reviewer 2 Report
Comments
The authors of this study, Aylmore et al. have summarized the original works on specialized surgical instruments for endoscope-assisted neurosurgery in their systematic review. Data collection and presentation have both statistical and qualitative evidences. The review of original works are also convincing, but they are ignoring the science of these critical scientific instruments.
#1. First of all, the manuscript is well written; however several improvements in academic languages and punctuations is expected. Example, unnecessary comma on the title, starting statements with numerical values and may other points can be made.
#2. The work presented here has a clear aim, focus, and direction. It appeared that the aim was to summarize only statistics, but what about science that affects safety, efficacy and usability. As a systematic review, potential readers will expect how the features: shape, size, functioning and the technological improvements in the instruments increases their applicability/popularity.
#3. The summary of available instruments is truly appreciable. At-least one or two statements of each type, defining their critical features that make them different/specialized from the standard neurosurgical instruments need to discuss on section 3.6.
#4. It would be nice if they made the words within each figure with the same font, size, and equally legible.
#5. The conclusion section needs to be rewritten that should also answer, why their work is important?
Overall, the concept is exciting; original research works are well reviewed, but improvements can be made as listed above
Author Response
Thank you for taking the time to review our submission and for your comments.
We have made the following revisions:
Reviewer comment: #1. First of all, the manuscript is well written; however several improvements in academic languages and punctuations is expected. Example, unnecessary comma on the title, starting statements with numerical values and may other points can be made.
Revisions made: The manuscript has been thoroughly proof-read for language and punctuation mistakes. All Oxford commas, including the one noted in the title, have been removed and all numerical values at the start of sentences have been removed. There are further minor changes throughout to spelling and punctuation that are marked by the tracked changes function.
Reviewer comment: #2. The work presented here has a clear aim, focus, and direction. It appeared that the aim was to summarize only statistics, but what about science that affects safety, efficacy and usability. As a systematic review, potential readers will expect how the features: shape, size, functioning and the technological improvements in the instruments increases their applicability/popularity.
Revisions made: To address “the science that affects safety, efficacy and usability” an additional column has been added to Table 1 entitled ‘Features’. This column contains all available information from each included article about the instruments features including shape, size, functioning and technological improvements.
Reviewer comment: #3. The summary of available instruments is truly appreciable. At-least one or two statements of each type, defining their critical features that make them different/specialized from the standard neurosurgical instruments need to discuss on section 3.6.
Revisions made: In addition to highlighting the critical features that make the specialised instruments different from standard instruments in Table 1, in Section 3.6 the differences have been made more apparent by the inclusion of more information regarding what type of standard instrument the novel specialised instrument was compared to in each study.
Reviewer comment: #4. It would be nice if they made the words within each figure with the same font, size, and equally legible.
Revisions made: Figure 2 has been remade in the style of Figures 3 and 4. The size of all Figures has been increased to improve legibility.
Reviewer comment: #5. The conclusion section needs to be rewritten that should also answer, why their work is important?
Revisions made: The conclusion section has been rewritten to further highlight why we believe this work is important.
If you have any other comments following these revisions we would be happy to address them.